# STABILIZING TRANSFORMERS FOR REINFORCEMENT LEARNING

## ABSTRACT

Owing to their ability to both effectively integrate information over long time horizons and scale to massive amounts of data, self-attention architectures have recently shown breakthrough success in natural language processing (NLP), achieving state-of-the-art results in domains such as language modeling and machine translation. Harnessing the transformer's ability to process long time horizons of information could provide a similar performance boost in partially observable reinforcement learning (RL) domains, but the large-scale transformers used in NLP have yet to be successfully applied to the RL setting. In this work we demonstrate that the standard transformer architecture is difficult to optimize, which was previously observed in the supervised learning setting but becomes especially pronounced with RL objectives. We propose architectural modifications that substantially improve the stability and learning speed of the original Transformer and XL variant. The proposed architecture, the Gated Transformer-XL (GTrXL), surpasses LSTMs on challenging memory environments and achieves state-of-the-art results on the multi-task DMLab-30 benchmark suite, exceeding the performance of an external memory architecture. We show that the GTrXL, trained using the same losses, has stability and performance that consistently matches or exceeds a competitive LSTM baseline, including on more reactive tasks where memory is less critical. GTrXL offers an easy-to-train, simple-to-implement but substantially more expressive architectural alternative to the standard multi-layer LSTM ubiquitously used for RL agents in partially observable environments.

## 1 INTRODUCTION

It has been argued that self-attention architectures (Vaswani et al., 2017) deal better with longer temporal horizons than recurrent neural networks (RNNs): by construction, they avoid compressing the whole past into a fixed-size hidden state and they do not suffer from vanishing or exploding gradients in the same way as RNNs. Recent work has empirically validated these claims, demonstrating that self-attention architectures can provide significant gains in performance over the more traditional recurrent architectures such as the LSTM (Dai et al., 2019; Radford et al., 2019; Devlin et al., 2019; Yang et al., 2019). In particular, the Transformer architecture (Vaswani et al., 2017) has had breakthrough success in a wide variety of domains: language modeling (Dai et al., 2019; Radford et al., 2019; Yang et al., 2019), machine translation (Vaswani et al., 2017; Edunov et al., 2018), summarization (Liu & Lapata), question answering (Dehghani et al., 2018; Yang et al., 2019), multi-task representation learning for NLP (Devlin et al., 2019; Radford et al., 2019; Yang et al., 2019), and algorithmic tasks (Dehghani et al., 2018).

The repeated success of the transformer architecture in domains where sequential information processing is critical to performance makes it an ideal candidate for partially observable RL problems, where episodes can extend to thousands of steps and the critical observations for any decision often span the entire episode. Yet, the RL literature is dominated by the use of LSTMs as the main mechanism for providing memory to the agent (Espeholt et al., 2018; Kapturowski et al., 2019; Mnih et al., 2016). Despite progress at designing more expressive memory architectures (Graves et al., 2016; Wayne et al., 2018; Santoro et al., 2018) that perform better than LSTMs in memory-based tasks and partially-observable environments, they have not seen widespread adoption in RL agents perhaps due to their complex implementation, with the LSTM being seen as the go-to solution for environments where memory is required. In contrast to these other memory architectures, the transformer

is well-tested in many challenging domains and has seen several open-source implementations in a variety of deep learning frameworks [1].

Motivated by the transformer's superior performance over LSTMs and the widespread availability of implementations, in this work we investigate the transformer architecture in the RL setting. In particular, we find that the canonical transformer is significantly difficult to optimize, often resulting in performance comparable to a random policy. This difficulty in training transformers exists in the supervised case as well. Typically a complex learning rate schedule is required (e.g., linear warmup or cosine decay) in order to train (Vaswani et al., 2017; Dai et al., 2019), or specialized weight initialization schemes are used to improve performance (Radford et al., 2019). These measures do not seem to be sufficient for RL. In Mishra et al. (2018), for example, transformers could not solve even simple bandit tasks and tabular Markov Decision Processes (MDPs), leading the authors to hypothesize that the transformer architecture was not suitable for processing sequential information.

However in this work we succeed in stabilizing training with a reordering of the layer normalization coupled with the addition of a new gating mechanism to key points in the submodules of the transformer. Our novel gated architecture, the Gated Transformer-XL (GTrXL) (shown in Figure 1, Right), is able to learn much faster and more reliably and exhibit significantly better final performance than the canonical transformer. We further demonstrate that the GTrXL achieves state-of-the-art results when compared to the external memory architecture MERLIN (Wayne et al., 2018) on the multitask DMLab-30 suite (Beattie et al., 2016). Additionally, we surpass LSTMs significantly on memory-based DMLab-30 levels while matching performance on the reactive set, as well as significantly outperforming LSTMs on memory-based continuous control and navigation environments. We perform extensive ablations on the GTrXL in challenging environments with both continuous actions and high-dimensional observations, testing the final performance of the various components as well as the GTrXL's robustness to seed and hyperparameter sensitivity compared to LSTMs and the canonical transformer. We demonstrate a consistent superior performance while matching the stability of LSTMs, providing evidence that the GTrXL architecture can function as a drop-in replacement to the LSTM networks ubiquitously used in RL.

## 2 TRANSFORMER ARCHITECTURE AND VARIANTS

The transformer network consists of several stacked blocks that repeatedly apply self-attention to the input sequence. The transformer layer block itself has remained relatively constant since its original introduction (Vaswani et al., 2017; Liu et al., 2018; Radford et al., 2019). Each layer consists of two submodules: an attention operation followed by a position-wise multi-layer network (see Figure 1 (left)). The input to the transformer block is an embedding from the previous layer $E^{(l-1)} \in \mathbb{R}^{T \times D}$, where $T$ is the number of time steps, $D$ is the hidden dimension, and $l \in [0, L]$ is the layer index with $L$ being the total number of layers. We assume $E^{(0)}$ is an arbitrarily-obtained input embedding of dimension $[T, D]$, e.g. a word embedding in the case of language modeling or an embedding of the per-timestep observations in an RL environment.

**Multi-Head Attention:** The Multi-Head Attention (MHA) submodule computes in parallel $H$ soft-attention operations for every time step. A residual connection (He et al., 2016a) and layer normalization (Ba et al., 2016) are then applied to the output (see Appendix C for more details):

$$\overline{Y}^{(l)} = \text{MultiHeadAttention}(E^{(l-1)}), \quad \hat{Y}^{(l)} = E^{(l-1)} + \overline{Y}^{(l)}, \quad Y^{(l)} = \text{LayerNorm}(\hat{Y}^{(l)}), \quad (1)$$

**Multi-Layer Perceptron:** The Multi-Layer Perceptron (MLP) submodule applies a $1 \times 1$ temporal convolutional network $f^{(l)}$ (i.e., kernel size 1, stride 1) over every step in the sequence, producing a new embedding tensor $E^{(l)} \in \mathbb{R}^{T \times D}$. As in Dai et al. (2019), the network output does not include an activation function. After the MLP, there is a residual update followed by layer normalization:

$$\overline{E}^{(l)} = f^{(l)}(Y^{(l)}), \qquad \hat{E}^{(l)} = Y^{(l)} + \overline{E}^{(l)}, \qquad E^{(l)} = \text{LayerNorm}(\hat{E}^{(l)}). \qquad (2)$$

**Relative Position Encodings:** The basic MHA operation does not take sequence order into account explicitly because it is permutation invariant. Positional encodings are a widely used solution in

---

[1]e.g. https://github.com/kimiyoung/transformer-xl, https://github.com/tensorflow/tensor2tensor

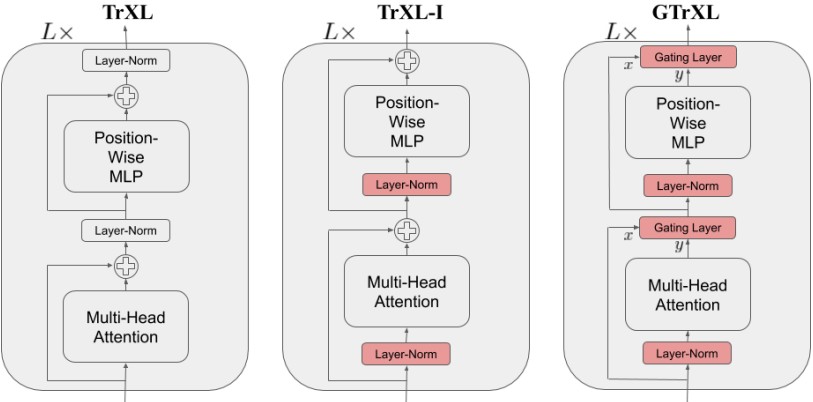

Figure 1: Transformer variants, showing just a single layer block (there are $L$ layers total). **Left:** Canonical Transformer(-XL) block with multi-head attention and position-wise MLP submodules and the standard layer normalization (Ba et al., 2016) placement with respect to the residual connection (He et al., 2016a). **Center:** TrXL-I moves the layer normalization to the input stream of the submodules. Coupled with the residual connections, there is a gradient path that flows from output to input without any transformations. **Right:** The GTrXL block, which additionally adds a gating layer in place of the residual connection of the TrXL-I.

domains like language where order is an important semantic cue, appearing in the original transformer architecture (Vaswani et al., 2017). To enable a much larger contextual horizon than would otherwise be possible, we use the relative position encodings and memory scheme used in Dai et al. (2019). In this setting, there is an additional $\mathcal{T}$-step memory tensor $M^{(l)} \in \mathbb{R}^{\mathcal{T} \times D}$, which is treated as constant during weight updates. The MHA submodule then becomes:

$$\overline{Y}^{(l)} = \text{RelativeMultiHeadAttention}(\text{StopGrad}(M^{(l-1)}), E^{(l-1)}) \tag{3}$$

$$\hat{Y}^{(l)} = E^{(l-1)} + \overline{Y}^{(l)}, \qquad Y^{(l)} = \text{LayerNorm}(\hat{Y}^{(l)}) \tag{4}$$

where StopGrad is a stop-gradient function that prevents gradients flowing backwards during backpropagation. We refer to Appendix C for a more detailed description.

## 3 GATED TRANSFORMER ARCHITECTURES

While the transformer architecture has achieved breakthrough results in modeling sequences for supervised learning tasks (Vaswani et al., 2017; Liu et al., 2018; Dai et al., 2019), a demonstration of the transformer as a useful RL memory has been notably absent. Previous work has highlighted training difficulties and poor performance (Mishra et al., 2018). When transformers have not been used for temporal memory but instead as a mechanism for attention over the input space, they have had success—notably in the challenging multi-agent Starcraft 2 environment (Vinyals et al., 2019). Here, the transformer was applied solely across Starcraft units and not over time.

Multiplicative interactions have been successful at stabilizing learning across a wide variety of architectures (Hochreiter & Schmidhuber, 1997; Srivastava et al., 2015; Cho et al., 2014). Motivated by this, we propose the introduction of powerful gating mechanisms in place of the residual connections within the transformer block, coupled with changes to the order of layer normalization in the submodules. As will be empirically demonstrated, the "Identity Map Reordering" and gating mechanisms are critical for stabilizing learning and improving performance.

### 3.1 IDENTITY MAP REORDERING

Our first change is to place the layer normalization on only the input stream of the submodules, a modification described in several previous works (He et al., 2016b; Radford et al., 2019; Baevski

& Auli, 2019). The model using this *Identity Map Reordering* is termed TrXL-I in the following, and is depicted visually in Figure 1 (center). A key benefit to this reordering is that it now enables an identity map from the input of the transformer at the first layer to the output of the transformer after the last layer. This is in contrast to the canonical transformer, where there are a series of layer normalization operations that non-linearly transform the state encoding. Because the layer norm reordering causes a path where two linear layers are applied in sequence, we apply a ReLU activation to each sub-module output before the residual connection (see Appendix C for equations).

The TrXL-I already exhibits a large improvement in stability and performance over TrXL (see Section 4.3.1). One hypothesis as to why the Identity Map Reordering improves results is as follows: assuming that the submodules at initialization produce values that are in expectation near zero, the state encoding is passed un-transformed to the policy and value heads, enabling the agent to learn a Markovian policy at the start of training (i.e., the network is initialized such that $\pi(\cdot|s_t, \ldots, s_1) \approx \pi(\cdot|s_t)$ and $V^\pi(s_t|s_{t-1}, \ldots, s_1) \approx V^\pi(s_t|s_{t-1})$). In many environments, reactive behaviours need to be learned before memory-based ones can be effectively utilized, i.e., an agent needs to learn how to walk before it can learn how to remember where it has walked.

### 3.2 GATING LAYERS

We further improve performance and optimization stability by replacing the residual connections in Equations 4 and 2 with gating layers. We call the gated architecture with the identity map reordering the *Gated Transformer(-XL)* (GTrXL). The final GTrXL layer block is written below:

$$\overline{Y}^{(l)} = \text{RelativeMultiHeadAttention}(\text{LayerNorm}([\text{StopGrad}(M^{(l-1)}), E^{(l-1)}])) \tag{5}$$

$$Y^{(l)} = g_{\text{MHA}}^{(l)}(E^{(l-1)}, \text{ReLU}(\overline{Y}^{(l)})) \tag{6}$$

$$\overline{E}^{(l)} = f^{(l)}(\text{LayerNorm}(Y^{(l)})) \tag{7}$$

$$E^{(l)} = g_{\text{MLP}}^{(l)}(Y^{(l)}, \text{ReLU}(\overline{E}^{(l)})) \tag{8}$$

where $g$ is a gating layer function. A visualization of our final architecture is shown in Figure 1 (right), with the modifications from the canonical transformer highlighted in red. In our experiments we ablate a variety of gating layers with increasing expressivity:

**Input:** The gated input connection has a sigmoid modulation on the input stream, similar to the short-cut-only gating from He et al. (2016b):

$$g^{(l)}(x, y) = \sigma(W_g^{(l)} x) \odot x + y$$

**Output:** The gated output connection has a sigmoid modulation on the output stream:

$$g^{(l)}(x, y) = x + \sigma(W_g^{(l)} x - b_g^{(l)}) \odot y$$

**Highway:** The highway connection (Srivastava et al., 2015) modulates both streams with a sigmoid:

$$g^{(l)}(x, y) = \sigma(W_g^{(l)} x + b_g^{(l)}) \odot x + (1 - \sigma(W_g^{(l)} x + b_g^{(l)})) \odot y$$

**Sigmoid-Tanh:** The sigmoid-tanh (SigTanh) gate (Van den Oord et al., 2016) is similar to the Output gate but with an additional tanh activation on the output stream:

$$g^{(l)}(x, y) = x + \sigma(W_g^{(l)} y - b) \odot \tanh(U_g^{(l)} y)$$

**Gated-Recurrent-Unit-type gating:** The Gated Recurrent Unit (GRU) (Chung et al., 2014) is a recurrent network that performs similarly to an LSTM (Hochreiter & Schmidhuber, 1997) but has fewer parameters. We adapt its powerful gating mechanism as an untied activation function in depth:

$$r = \sigma(W_r^{(l)} y + U_r^{(l)} x), \qquad z = \sigma(W_z^{(l)} y + U_z^{(l)} x - b_g^{(l)}), \qquad \hat{h} = \tanh(W_g^{(l)} y + U_g^{(l)}(r \odot x)),$$

$$g^{(l)}(x, y) = (1 - z) \odot x + z \odot \hat{h}.$$

**Gated Identity Initialization:** We have claimed that the Identity Map Reordering aids policy optimization because it initializes the agent close to a Markovian policy / value function. If this is indeed the cause of improved stability, we can explicitly initialize the various gating mechanisms to be close to the identity map. This is the purpose of the bias $b_g^{(l)}$ in the applicable gating layers. We later demonstrate in an ablation that initially setting $b_g^{(l)} > 0$ can greatly improve learning speed.

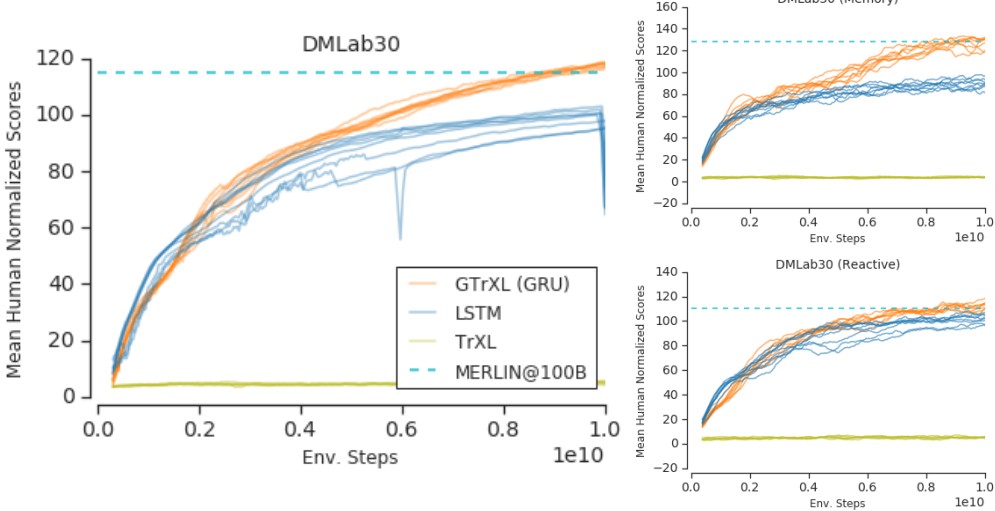

Figure 2: Average return on DMLab-30, re-scaled such that a human has mean 100 score on each level and a random policy has 0. **Left:** Results averaged over the full DMLab-30 suite. **Right:** DMLab-30 partitioned into a "Memory" and "Reactive" split (described in Appendix D). The GTrXL has a substantial gain over LSTM in memory-based environments, while even slightly surpassing performance on the reactive set. We plot 6-8 hyperparameter settings per architecture (see Appendix B). MERLIN scores obtained from personal communication with the authors.

## 4 EXPERIMENTS

In this section, we provide experiments on a variety of challenging single and multi-task RL domains: DMLab-30 (Beattie et al., 2016), Numpad and Memory Maze (see Fig. 8). Crucially we demonstrate that the proposed Gated Transformer-XL (GTrXL) not only shows substantial improvements over LSTMs on memory-based environments, but suffers no degradation of performance on reactive environments. The GTrXL also exceeds MERLIN (Wayne et al., 2018), an external memory architecture which used a Differentiable Neural Computer (Graves et al., 2016) coupled with auxiliary losses, surpassing its performance on both memory and reactive tasks.

For all transformer architectures except when otherwise stated, we train relatively deep 12-layer networks with embedding size 256 and memory size 512. These networks are comparable to the state-of-the-art networks in use for small language modeling datasets (see enwik8 results in (Dai et al., 2019)). We chose to train deep networks in order to demonstrate that our results do not necessarily sacrifice complexity for stability, i.e. we are not making transformers stable for RL simply by making them shallow. Our networks have receptive fields that can potentially span any episode in the environments tested, with an upper bound on the receptive field of 6144 (12 layers $\times$ 512 memory (Dai et al., 2019)). Future work will look at scaling transformers in RL even further, e.g. towards the 52-layer network in Radford et al. (2019). See App. B for experimental details.

For all experiments, we used V-MPO (Anonymous Authors, 2019), an on-policy adaptation of Maximum a Posteriori Policy Optimization (MPO) (Abdolmaleki et al., 2018a;b) that performs approximate policy iteration based on a learned state-value function $V(s)$ instead of the state-action value function used in MPO. Rather than directly updating the parameters in the direction of the policy gradient, V-MPO uses the estimated advantages to first construct a target distribution for the policy update subject to a sample-based KL constraint, then calculates the gradient that partially moves the parameters toward that target, again subject to a KL constraint. V-MPO was shown to achieve state-of-the-art results for LSTM-based agents on the multi-task DMLab-30 benchmark suite.

### 4.1 TRANSFORMER AS EFFECTIVE RL MEMORY ARCHITECTURE

We first present results of the best performing GTrXL variant, the GRU-type gating, against a competitive LSTM baseline, demonstrating a substantial improvement on the multi-task DMLab-30 do-

| Model | Mean Human Norm. | Mean Human Norm., 100-capped |
|---|---|---|
| LSTM | $99.3 \pm 1.0$ | $84.0 \pm 0.4$ |
| TrXL | $5.0 \pm 0.2$ | $5.0 \pm 0.2$ |
| TrXL-I | $107.0 \pm 1.2$ | $87.4 \pm 0.3$ |
| MERLIN@100B | 115.2 | 89.4 |
| GTrXL (GRU) | $117.6 \pm 0.3$ | $89.1 \pm 0.2$ |
| GTrXL (Input) | $51.2 \pm 13.2$ | $47.6 \pm 12.1$ |
| GTrXL (Output) | $112.8 \pm 0.8$ | $87.8 \pm 0.3$ |
| GTrXL (Highway) | $90.9 \pm 12.9$ | $75.2 \pm 10.4$ |
| GTrXL (SigTanh) | $101.0 \pm 1.3$ | $83.9 \pm 0.7$ |

Table 1: Final human-normalized return averaged across all 30 DMLab levels for baselines and GTrXL variants. We also include the 100-capped score where the per-level mean score is clipped at 100, providing a metric that is proportional to the percentage of levels that the agent is superhuman. We see that the GTrXL (GRU) surpasses LSTM by a significant gap and exceeds the performance of MERLIN (Wayne et al., 2018) trained for 100 billion environment steps. The GTrXL (Output) and the proposed reordered TrXL-I also surpass LSTM but perform slightly worse than MERLIN and are not as robust as GTrXL (GRU) (see Sec. 4.3.2). We sample 6-8 hyperparameters per model. We include standard error over runs.

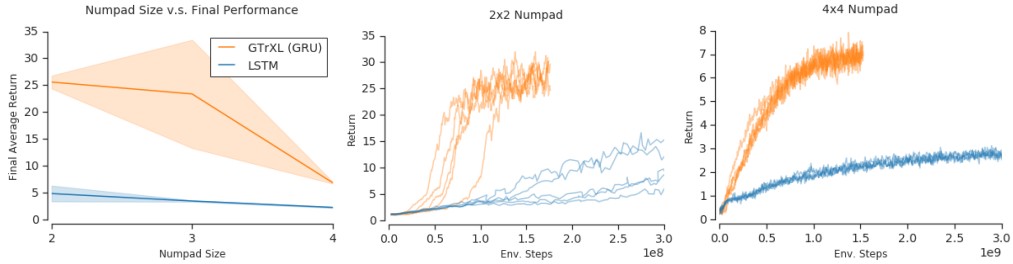

Figure 3: Numpad results demonstrating that the GTrXL has much better memory scaling properties than LSTM. **Left:** As the Numpad environment's memory requirement increases (because of larger pad size), the GTrXL suffers much less than LSTM. However, because of the combinatorial nature of Numpad, the GTrXL eventually also starts dropping in performance at 4x4. We plot mean and standard error of the last 200 episodes after training each model for 0.15B, 1.0B and 2.0B environment steps for Numpad size 2, 3 and 4, respectively. **Center, Right:** Learning curves for the GTrXL on $2 \times 2$ and $4 \times 4$ Numpad. Even when the LSTM is trained for twice as long, the GTrXL still has a substantial improvement over it. We plot 5 hyperparameter settings per model for learning curves.

main (Beattie et al., 2016). DMLab-30 is a large-scale, multitask benchmark comprising 30 first-person 3D environments with image observations and has been widely used as a benchmark for architectural and algorithmic improvements (Wayne et al., 2018; Espeholt et al., 2018; Kapturowski et al., 2019; Hessel et al., 2018). The levels test a wide agent competencies such as language comprehension, navigation, handling of partial observability, memory, planning, and other forms of long horizon reasoning, with episodes lasting over 4000 environment steps. Figure 2 shows mean return over all levels as training progresses, where the return is human normalized as done in previous work (meaning a human has a per-level mean score of 100 and a random policy has a score of 0), while Table 1 has the final performance at 10 billion environment steps. The GTrXL has a significant gap over a 3-layer LSTM baseline trained using the same V-MPO algorithm. Furthermore, we included the final results of a previously-published external memory architecture, MERLIN (Wayne et al., 2018). Because MERLIN was trained for 100 billion environment steps with a different algorithm, IMPALA (Espeholt et al., 2018), and also involved an auxiliary loss critical for the memory component to function, the learning curves are not directly comparable and we only report the final performance of the architecture as a dotted line. Despite the differences, our results demonstrate that the GTrXL can match the previous state-of-the-art on DMLab-30. An informative split between a set of memory-based levels and more reactive ones (listed in Appendix D) reveals that our model specifically has large improvements in environments where memory plays a critical role. Meanwhile, GTrXL also shows improvement over LSTMs on the set of reactive levels, as memory can still be effectively utilized in some of these levels.

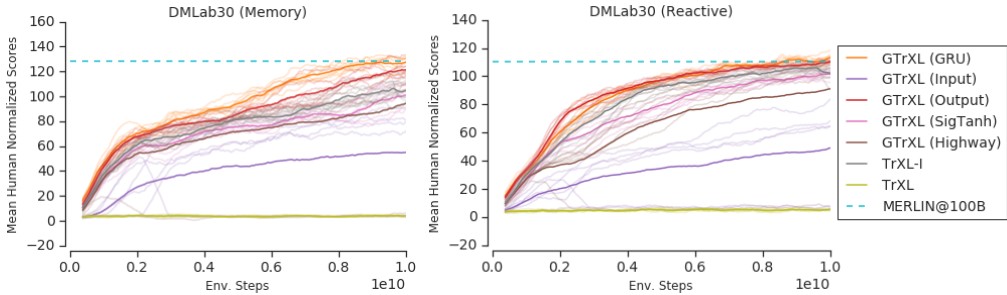

Figure 4: Learning curves for the gating mechanisms, along with MERLIN score at 100 billion frames as a reference point. We can see that the GRU performs as well as any other gating mechanism on the reactive set of tasks. On the memory environments, the GRU gating has a significant gain in learning speed and attains the highest final performance at the fastest rate. We plot both mean (bold) and the individual 6-8 hyperparameter samples per model (light).

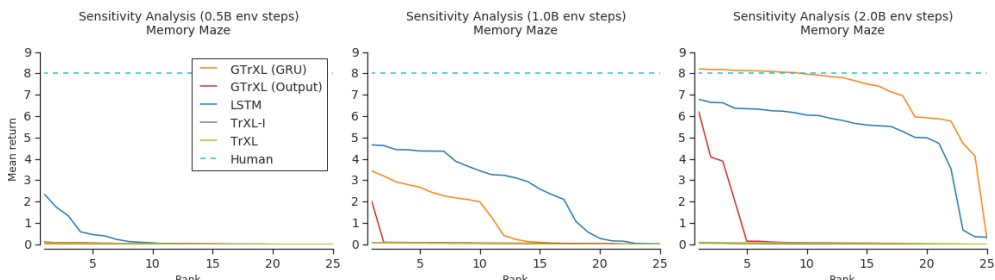

Figure 5: Sensitivity analysis of GTrXL variants versus TrXL and LSTM baselines. We sample 25 different hyperparameter sets and seeds and plot the ranked average return at 3 points during training (0.5B, 1.0B and 2.0B environment steps). Higher and flatter lines indicate more robust architectures. The same hyperparameter sampling distributions were used across models (see Appendix B). We plot human performance as a dotted line.

## 4.2 SCALING WITH MEMORY HORIZON

We next demonstrate that the GTrXL scales better compared to an LSTM when an environment's temporal horizon is increased, using the "Numpad" continuous control task of Humplik et al. (2019) which allows an easy combinatorial increase in the temporal horizon. In Numpad, a robotic agent is situated on a platform resembling the 3x3 number pad of a telephone (generalizable to $N \times N$ pads). The agent can interact with the pads by colliding with them, causing them to be activated (visualized in the environment state as the number pad glowing). The goal of the agent is to activate a specific sequence of up to $N^2$ numbers, but without knowing this sequence a priori. The only feedback the agent gets is by activating numbers: if the pad is the next one in the sequence, the agent gains a reward of +1, otherwise all activated pads are cleared and the agent must restart the sequence. Each correct number in the sequence only provides reward once, i.e. each subsequent activation of that number will no longer provide rewards. Therefore the agent must explicitly develop a search strategy to determine the correct pad sequence. Once the agent completes the full sequence, all pads are reset and the agent gets a chance to repeat the sequence again for more reward. This means higher reward directly translates into how well the pad sequence has been memorized. An image of the scenario is provided in Figure 3. There is the restriction that contiguous pads in the sequence must be contiguous in space, i.e. the next pad in the sequence can only be in the Moore neighborhood of the previous pad. Furthermore, no pad can be pressed twice in the sequence.

We present two results in this environment in Figure 3. The first measures the final performance of the trained models as a function of the pad size. We can see that LSTM performs badly on all 3 pad sizes, and performs worse as the pad size increases from 2 to 4. The GTrXL performs much better, and almost instantly solves the environment with its much more expressive memory. On the center

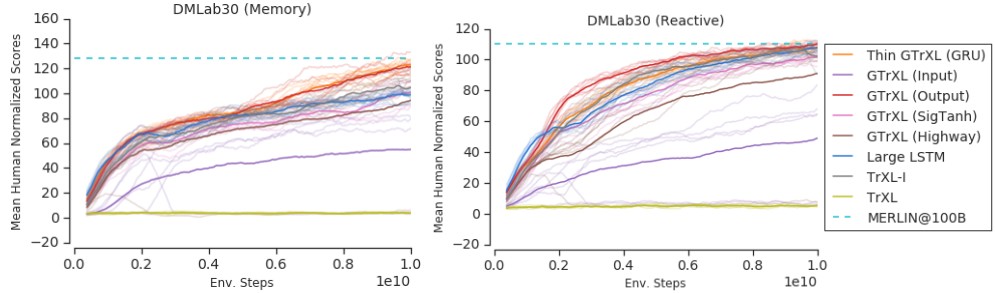

Figure 6: Learning curves comparing a thinner GTrXL (GRU) with half the embedding dimension of the other presented gated variants and TrXL baselines. The Thin GTrXL (GRU) has fewer parameters than any other model presented but still matches the performance of the best performing counterpart, the GTrXL (Output), which has over 10 million more parameters. We plot both mean (bold) and 6-8 hyperparameter settings (light) per model.

| Model | Mean Human Norm. Score | # Param. Millions |
|---|---|---|
| LSTM | $99.3 \pm 1.0$ | 9.25M |
| Large LSTM | $103.5 \pm 0.9$ | 51.3M |
| TrXL | $5.0 \pm 0.2$ | 28.6M |
| TrXL-I | $107.0 \pm 1.2$ | 28.6M |
| Thin GTrXL (GRU) | $111.5 \pm 0.6$ | 22.4M |
| GTrXL (GRU) | $117.6 \pm 0.3$ | 66.4M |
| GTrXL (Input) | $51.2 \pm 13.2$ | 34.9M |
| GTrXL (Output) | $112.8 \pm 0.8$ | 34.9M |
| GTrXL (Highway) | $90.9 \pm 12.9$ | 34.9M |
| GTrXL (SigTanh) | $101.0 \pm 1.3$ | 41.2M |

Table 2: Parameter-controlled ablation. We report standard error of the means of 6-8 runs per model.

| Model | % Diverged |
|---|---|
| LSTM | 0% |
| TrXL | 0% |
| TrXL-I | 16% |
| GTrXL (GRU) | 0% |
| GTrXL (Output) | 12% |

Table 3: Percentage of the 25 parameter settings where the training loss diverged within 2 billion env. steps. We do not report numbers for GTrXL gating types that were unstable in DMLab-30. For diverged runs we plot the returns in Figure 5 as 0 afterwards.

and right images, we provide learning curves for the $2 \times 2$ and $4 \times 4$ Numpad environments, and show that even when the LSTM is trained twice as long it does not reach GTrXL's performance.

### 4.3 GATING VARIANTS + IDENTITY MAP REORDERING

We demonstrated that the GRU-type-gated GTrXL can achieve state-of-the-art results on DMLab-30, surpassing both a deep LSTM and an external memory architecture, and also that the GTrXL has a memory which scales better with the memory horizon of the environment. However, the question remains whether the expressive gating mechanisms of the GRU could be replaced by simpler alternatives. In this section, we perform extensive ablations on the gating variants described in Section 3.2, and show that the GTrXL (GRU) has improvements in learning speed, final performance and optimization stability over all other models, even when controlling for the number of parameters.

#### 4.3.1 PERFORMANCE ABLATION

We first report the performance of the gating variants in DMLab-30. Table 1 and Figure 4 show the final performance and training curves of the various gating types in both the memory / reactive split, respectively. The canonical TrXL completely fails to learn, while the TrXL-I improves over the LSTM. Of the gating varieties, the GTrXL (Output) can recover a large amount of the performance of the GTrXL (GRU), especially in the reactive set, but as shown in Sec. 4.3.2 is generally far less stable. The GTrXL (Input) performs worse than even the TrXL-I, reinforcing the identity map path hypothesis. Finally, the GTrXL (Highway) and GTrXL (SigTanh) are more sensitive to the hyperparameter settings compared to the alternatives, with some settings doing worse than TrXL-I.

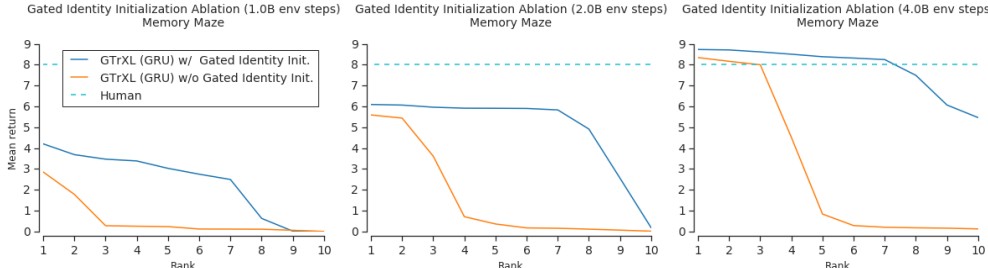

Figure 7: Ablation of the gated identity initialization on Memory Maze by comparing 10 runs of a model run with the bias initialization and 10 runs of a model without. Every run has independently sampled hyperparameters from a distribution. We plot the ranked mean return of the 10 runs of each model at 1, 2, and 4 billion environment steps. Each mean return is the average of the past 200 episodes at the point of the model snapshot. We plot human performance as a dotted line.

### 4.3.2 HYPERPARAMETER AND SEED SENSITIVITY

Beyond improved performance, we next demonstrate a significant reduction in hyperparameter and seed sensitivity for the GTrXL (GRU) compared to baselines and other GTrXL variants. We use the "Memory Maze" environment, a memory-based navigation task in which the agent must discover the location of an apple randomly placed in a maze of blocks. The agent receives a positive reward for collecting the apple and is then teleported to a random location in the maze, with the apple's position held fixed. The agent can make use of landmarks situated around the room to return as quickly as possible to the apple for subsequent rewards. Therefore, an effective mapping of the environment results in more frequent returns to the apple and higher reward.

We chose to perform the sensitivity ablation on Memory Maze because (1) it requires the use of long-range memory to be effective and (2) it includes both continuous and discrete action sets (details in Appendix A) which makes optimization more difficult. In Figure 5, we sample 25 independent V-MPO hyperparameter settings from a wide range of values and train the networks to 2 billion environment steps (see Appendix B). Then, at various points in training (0.5B, 1.0B and 2.0B), we rank all runs by their mean return and plot this ranking. Models with curves which are both higher and flatter are thus more robust to hyperparameters and random seeds. Our results demonstrate that (1) the GTrXL (GRU) can learn this challenging memory environment in much fewer environment steps than LSTM, and (2) that GTrXL (GRU) beats the other gating variants in stability by a large margin, thereby offering a substantial reduction in necessary hyperparameter tuning. The values in Table 3 list what percentage of the 25 runs per model had losses that diverged to infinity. We can see that the only model reaching human performance in 2 billion environment steps is the GTrXL (GRU), with 10 runs having a mean score 8 and above.

### 4.3.3 PARAMETER COUNT-CONTROLLED COMPARISONS

For the final gating ablation, we compare transformer variants while tracking their total parameter count to control for the increase in capacity caused by the introduction of additional parameters in the gating mechanisms. To demonstrate that the advantages of the GTrXL (GRU) are not solely due to an increase in parameter count, we halve the number of attention heads (which also effectively halves the embedding dimension due to the convention that the embedding size is the number of heads multiplied by the attention head dimension). The effect is a substantial reduction in parameter count, resulting in less parameters than even the canonical TrXL. Fig. 6 and Tab. 2 compare the different models to the "Thin" GTrXL (GRU), with Tab. 2 listing the parameter counts. We include a parameter-matched LSTM model with 12 layers and 512 hidden size. The Thin GTrXL (GRU) surpasses every other model (within variance) except the GTrXL (GRU), even surpassing the next best-performing model, the GTrXL (Output), with over 10 million less parameters.

### 4.3.4 GATED IDENTITY INITIALIZATION ABLATION

All applicable gating variants in the previous sections were trained with the gated identity initialization. We observed in initial Memory Maze results that the gated identity initialization significantly

improved optimization stability and learning speed. Figure 7 compares an otherwise identical 4-layer GTrXL (GRU) trained with and without the gated identity initialization. Similarly to the previous sensitivity plots, we plot the ranked mean return of 10 runs at various times during training. As can be seen from Fig. 7, there is a significant gap caused by the bias initialization, suggesting that preconditioning the transformer to be close to Markovian results in large learning speed gains.

## 5 RELATED WORK

Gating has been shown to be effective to address the vanishing gradient problem and thus improve the learnability of recurrent models. LSTM networks (Hochreiter & Schmidhuber, 1997; Graves, 2013) rely on an input, forget and output gate that protect the update of the cell. GRU (Chung et al., 2014; Cho et al., 2014) is another popular gated recurrent architecture that simplifies the LSTM cell, reducing the number of gates to two. Finding an optimal gating mechanism remains an active area of research, with other existing proposals (Krause et al., 2016; Kalchbrenner et al., 2015; Wu et al., 2016), as well as works trying to discover optimal gating by neural architecture search (Zoph & Le, 2017) More generally, gating and multiplicative interactions have a long history (Rumelhart et al., 1986). Gating has been investigated previously for improving the representational power of feedforward and recurrent models (Van den Oord et al., 2016; Dauphin et al., 2017), as well as learnability (Srivastava et al., 2015; Zilly et al., 2017). Initialization has also played a crucial role in making deep models trainable (LeCun et al., 1998; Glorot & Bengio, 2010; Sutskever et al., 2013).

There has been a wide variety of work looking at improving memory in reinforcement learning agents. External memory approaches typically have a regular feedforward or recurrent policy interact with a memory database through read and write operations. Priors are induced through the design of the specific read/write operations, such as those resembling a digital computer (Wayne et al., 2018; Graves et al., 2016) or an environment map (Parisotto & Salakhutdinov, 2018; Gupta et al., 2017). An alternative non-parametric perspective to memory stores an entire replay buffer of the agent's past observations, which is made available for the agent to itself reason over either through fixed rules (Blundell et al., 2016) or an attention operation (Pritzel et al., 2017). Others have looked at improving performance of LSTM agents by extending the architecture with stacked hierarchical connections / multiple temporal scales and auxiliary losses (Jaderberg et al., 2019; Stooke et al., 2019) or allowing an inner-loop update to the RNN weights (Miconi et al., 2018). Other work has examined self-attention in the context of exploiting relational structure within the input-space (Zambaldi et al., 2019) or within recurrent memories (Santoro et al., 2018).

## 6 CONCLUSION

In this paper we provided evidence that confirms previous observations in the literature that standard transformer models are unstable to train in the RL setting and often fail to learn completely (Mishra et al., 2018). We presented a new architectural variant of the transformer model, the GTrXL, which has increased performance, more stable optimization, and greater robustness to initial seed and hyperparameters than the canonical architecture. The key contributions of the GTrXL are reordered layer normalization modules and a gating layer instead of the standard residual connection. We performed extensive ablation experiments testing the robustness, ease of optimization and final performance of the gating layer variations, as well as the effect of the reordered layer normalization. These results empirically demonstrate that the GRU-type gating performs best across all metrics, exhibiting comparable robustness to hyperparameters and random seeds as an LSTM while still maintaining a performance improvement. Furthermore, the GTrXL (GRU) learns faster, more stably and achieves a higher final performance (even when controlled for parameters) than the other gating variants on the challenging multitask DMLab-30 benchmark suite.

Having demonstrated substantial and consistent improvement in DMLab-30, Numpad and Memory Maze over the ubiquitous LSTM architectures currently in use, the GTrXL makes the case for wider adoption of transformers in RL. A core benefit of the transformer architecture is its ability to scale to very large and deep models, and to effectively utilize this additional capacity in larger datasets. In future work, we hope to test the limits of the GTrXL's ability to scale in the RL setting by providing it with a large and varied set of training environments.

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

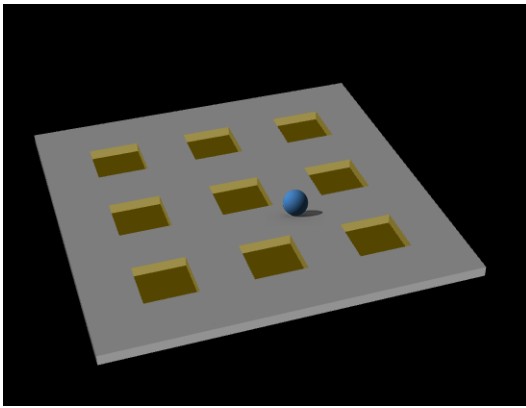 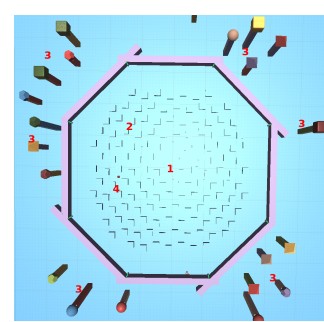

Figure 8: **Left:** The Numpad environment, showing the controllable "sphere" robot and a full 3x3 pad. Pads are activated when the robot collides with their center. The robot can move on the plane as well as jump to avoid pressing numbers. **Right:** Top down view of "Memory Maze": (1) Central chamber, (2) blocks among which the apple is placed, (3) landmarks the agent can use to locate the apple, (4) one of the possible location of the apple.

## APPENDIX

## A  ENVIRONMENT DETAILS

**Numpad:** Numpad has three actions, two of which move the sphere towards some direction in the x,y plane and the third allows the agent to jump in order to get over a pad faster. The observation consists of a variety of proprioceptive information (e.g. position, velocity, acceleration) as well as which pads in the sequence have been correctly activated (these will shut off if an incorrect pad is later hit), and the previous action and reward. Episodes last a fixed 500 steps and the agent can repeat the correct sequence any number of times to receive reward. Observations were processed using a simple 2-layer MLP with tanh activations to produce the transformer's input embedding.

**DMLab-30:** Ignoring the "jump" and "crouch" actions which we do not use, an action in the native DMLab action space consists of 5 integers whose meaning and allowed values are given in Table 4. Following previous work on DMLab (Hessel et al., 2018), we used the reduced action set given in Table 5 with an action repeat of 4. Observations are $72 \times 96$ RGB images. Some levels require a language input, and for that all models use an additional 64-dimension LSTM to process the sentence.

In Wayne et al. (2018), the DMLab Arbitrary Visuomotor Mapping task was specifically used to highlight the MERLIN architecture's ability to utilize memory. In Figure 9 we show that, given a similarly reduced action set as used in Wayne et al. (2018), see Table 6, the GTrXL architecture can also reliably attain human-level performance on this task.

| ACTION NAME | RANGE |
|---|---|
| LOOK_LEFT_RIGHT_PIXELS_PER_FRAME | [-512, 512] |
| LOOK_DOWN_UP_PIXELS_PER_FRAME | [-512, 512] |
| STRAFE_LEFT_RIGHT | [-1, 1] |
| MOVE_BACK_FORWARD | [-1, 1] |
| FIRE | [0, 1] |

Table 4: Native action space for DMLab. See `https://github.com/deepmind/lab/blob/master/docs/users/actions.md` for more details.

**Memory Maze:** An action in the native Memory Maze action space consists of 8 continuous actions and a single discrete action whose meaning and allowed values are given in Table 7. Unlike for DMLab, we used a hybrid continuous-discrete distribution (Neunert et al., 2019) to directly output policies in the game's native action space. Observations are $72 \times 96$ RGB images.

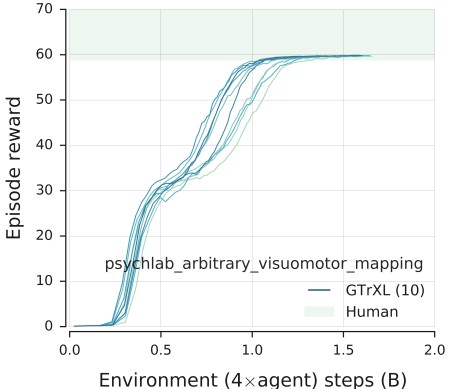

Figure 9: Learning curves for the DMLab Arbitrary Visuomotor Mapping task using a reduced action set.

| ACTION | NATIVE DMLAB ACTION |
|---|---|
| Forward (FW) | [ 0,    0,    0,    1,   0] |
| Backward (BW) | [ 0,    0,    0,   -1,   0] |
| Strafe left | [ 0,    0,   -1,    0,   0] |
| Strafe right | [ 0,    0,    1,    0,   0] |
| Small look left (LL) | [-10,    0,    0,    0,   0] |
| Small look right (LR) | [ 10,    0,    0,    0,   0] |
| Large look left (LL ) | [-60,    0,    0,    0,   0] |
| Large look right (LR) | [ 60,    0,    0,    0,   0] |
| Look down | [ 0,   10,    0,    0,   0] |
| Look up | [ 0,  -10,    0,    0,   0] |
| FW + small LL | [-10,    0,    0,    1,   0] |
| FW + small LR | [ 10,    0,    0,    1,   0] |
| FW + large LL | [-60,    0,    0,    1,   0] |
| FW + large LR | [ 60,    0,    0,    1,   0] |
| Fire | [ 0,    0,    0,    0,   1] |

Table 5: Simplified action set for DMLab from Hessel et al. (2018).

| ACTION | NATIVE DMLAB ACTION |
|---|---|
| Small look left (LL) | [-10,    0,    0,    0,   0] |
| Small look right (LR) | [ 10,    0,    0,    0,   0] |
| Look down | [ 0,   10,    0,    0,   0] |
| Look up | [ 0,  -10,    0,    0,   0] |
| No-op | [ 0,    0,    0,    0,   0] |

Table 6: Simplified action set for DMLab Arbitrary Visuomotor Mapping (AVM). This action set is the same as the one used for AVM in Wayne et al. (2018) but with an additional no-op, which may also be replaced with the *Fire* action.

**Image Encoder:** For DMLab-30 and Memory Maze, we used the same image encoder as in (Anonymous Authors, 2019) for multitask DMLab-30. The ResNet was adapted from Hessel et al. (2018) and each of its layer blocks consists of a $(3 \times 3,$ stride 1) convolution, followed by $(3 \times 3,$ stride 2) max-pooling, followed by 2 $3 \times 3$ residual blocks with ReLU non-linearities.

| ACTION NAME | RANGE |
|---|---|
| LOOK_LEFT_RIGHT | [-1.0, 1.0] |
| LOOK_DOWN_UP | [-1.0, 1.0] |
| STRAFE_LEFT_RIGHT | [-1.0, 1.0] |
| MOVE_BACK_FORWARD | [-1.0, 1.0] |
| HAND_ROTATE_AROUND_RIGHT | [-1.0, 1.0] |
| HAND_ROTATE_AROUND_UP | [-1.0, 1.0] |
| HAND_ROTATE_AROUND_FORWARD | [-1.0, 1.0] |
| HAND_PUSH_PULL | [-10.0, 10.0] |
| HAND_GRIP | {0, 1} |

Table 7: Hybrid action set for Memory Maze, consisting of 8 continuous actions and a single discrete action.

**Agent Output:** As in (Anonymous Authors, 2019), in all cases we use a 256-unit MLP with a linear output to get the policy logits (for discrete actions), Gaussian distribution parameters (for continuous actions) or value function estimates.

# B  EXPERIMENTAL DETAILS

For all experiments, beyond sampling independent random seeds, each run also has V-MPO hyper-parameters sampled from a distribution (see Table 8). The sampled hyperparameters are kept fixed across all models for a specific experiment, meaning that if one of the $\epsilon_\alpha$ sampled is 0.002, then all models will have 1 run with $\epsilon_\alpha = 0.002$ and so on for the rest of the samples. The exception is for the DMLab-30 LSTM, where a more constrained range was found to perform better in preliminary experiments. Each model had 8 seeds started, but not all runs ran to completion due to compute issues. These hyperparameter settings were dropped randomly and not due to poor environment performance. We report how many seeds ran to completion for all models. At least 6 seeds finished for every model tested. We list architecture details by section below. All LSTM models have residual skip connections in depth.

| Hyperparameter | Environment | | |
|---|---|---|---|
| | DMLab-30 | Numpad | Memory Maze |
| Batch Size | 128 | 128 | 128 |
| Unroll Length | 95 | 95 | 95 |
| Discount | 0.99 | 0.99 | 0.99 |
| Action Repeat | 4 | 1 | 4 |
| Pixel Control Cost | $2 \times 10^{-3}$ | - | - |
| Target Update Period | 10 | 10 | 10 |
| Initial $\eta$ | 1.0 | 10.0 | 1.0 |
| Initial $\alpha$ | 5.0 | - | 5.0 |
| Initial $\alpha_\mu$ | - | 1.0 | 1.0 |
| Initial $\alpha_\Sigma$ | - | 1.0 | 1.0 |
| $\epsilon_\eta$ | 0.1 | 0.1 | 0.1 |
| $\epsilon_\alpha$ (log-uniform) | LSTM [0.001, 0.025) TrXL Variants [0.001, 0.1) | - | [0.001, 0.1) |
| $\epsilon_{\alpha_\mu}$ (log-uniform) | - | [0.005, 0.01) | [0.005, 0.01) |
| $\epsilon_{\alpha_\Sigma}$ (log-uniform) | - | $[5 \times 10^{-6}, 4 \times 10^{-4})$ | $[5 \times 10^{-6}, 4 \times 10^{-5})$ |

Table 8: V-MPO hyperparameters per environment.

## B.1  TRAINING SETUP

All experiments in this work were carried out in an actor-learner framework (Espeholt et al., 2018) that utilizes TF-Replicator (Buchlovsky et al., 2019) for distributed training on TPUs in the 16-core

| Model | # Layers | Head Dim. | # Heads | Hidden Dim. | Memory Size | Runs Completed |
|---|---|---|---|---|---|---|
| LSTM | 3 | - | - | 256 | - | 8 |
| Large LSTM | 12 | - | - | 512 | - | 6 |
| TrXL | 12 | 64 | 8 | 512 | 512 | 6 |
| TrXL-I | 12 | 64 | 8 | 512 | 512 | 6 |
| GTrXL (GRU) | 12 | 64 | 8 | 512 | 512 | 8 |
| GTrXL (Input) | 12 | 64 | 8 | 512 | 512 | 6 |
| GTrXL (Output) | 12 | 64 | 8 | 512 | 512 | 7 |
| GTrXL (Highway) | 12 | 64 | 8 | 512 | 512 | 7 |
| GTrXL (SigTanh) | 12 | 64 | 8 | 512 | 512 | 6 |
| Thin GTrXL (GRU) | 12 | 64 | 4 | 256 | 512 | 8 |

Table 9: DMLab-30 Ablation Architecture Details. We report the number of runs per model that ran to completion (i.e. 10 billion environment steps). We follow the standard convention that the hidden/embedding dimension of transformers is equal to the head dimension multiplied by the number of heads. (Sec. 4.1 & Sec. 4.3).

| Model | # Layers | Head Dim. | # Heads | Hidden Dim. | Memory Size | Runs Completed |
|---|---|---|---|---|---|---|
| LSTM | 3 | - | - | 256 | - | 5 |
| GTrXL (GRU) | 12 | 64 | 8 | 256 | 512 | 5 |

Table 10: Numpad Architecture Details. (Sec. 4.2).

| Model | # Layers | Head Dim. | # Heads | Hidden Dim. | Memory Size |
|---|---|---|---|---|---|
| LSTM | 3 | - | - | 256 | - |
| TrXL | 12 | 64 | 8 | 256 | 512 |
| TrXL-I | 12 | 64 | 8 | 256 | 512 |
| GTrXL (GRU) | 12 | 64 | 8 | 256 | 512 |
| GTrXL (Output) | 12 | 64 | 8 | 256 | 512 |

Table 11: Sensitivity ablation architecture details (Sec. 4.3.2).

| Model | # Layers | Head Dim. | # Heads | Hidden Dim. | Memory Size | Runs Completed |
|---|---|---|---|---|---|---|
| GTrXL (GRU) | 4 | 64 | 4 | 256 | 512 | 8 |

Table 12: Gated identity initialization ablation architecture details (Sec. 4.3.4).

configuration (Google, 2018). "Actors" running on CPUs performed network inference and interactions with the environment, and transmitted the resulting trajectories to the centralised "learner".

## C  MULTI-HEAD ATTENTION DETAILS

### C.1  MULTI-HEAD ATTENTION

The Multi-Head Attention (MHA) submodule computes in parallel $H$ soft-attention operations for every time step, producing an output tensor $Y^{(l)} \in \mathbb{R}^{T \times D}$. MHA operates by first calculating the query $Q^{(l)} \in \mathbb{R}^{H \times T \times d}$, keys $K^{(l)} \in \mathbb{R}^{H \times T \times d}$, and values $V^{(l)} \in \mathbb{R}^{H \times T \times d}$ (where $d = D/H$) through trainable linear projections $W_Q^{(l)}$, $W_K^{(l)}$, and $W_V^{(l)}$, respectively, and then using the combined $Q$, $K$, $V$, tensors to compute the soft attention. A residual connection (He et al., 2016a) to the resulting embedding $E^{(l)}$ is then applied and finally layer normalization (Ba et al., 2016).

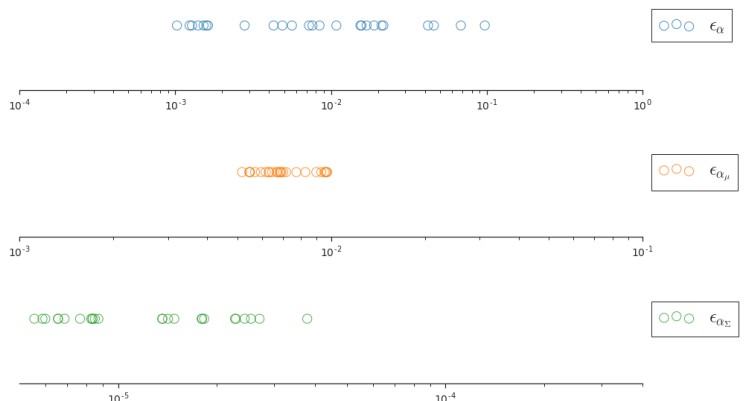

Figure 10: The 25 hyperparameter settings sampled for the sensitivity ablation (Sec. 4.3.2). X-axis is in log scale and values are sampled from the corresponding ranges given in Table 8.

**MultiHeadAttention**$(E^{(l-1)})$:

$$Q^{(l)}, K^{(l)}, V^{(l)} = W_Q^{(l)} E^{(l-1)}, W_K^{(l)} E^{(l-1)}, W_V^{(l)} E^{(l-1)} \tag{9}$$

$$\alpha_{htm}^{(l)} = Q_{htd} K_{hmd} \tag{10}$$

$$W_{htm}^{(l)} = \text{MaskedSoftmax}(\alpha^{(l)}, \text{axis}=m) \tag{11}$$

$$\overline{Y}_{htd}^{(l)} = W_{htm}^{(l)} V_{hmd}^{(l)} \tag{12}$$

$$\hat{Y}^{(l)} = E^{(l-1)} + \text{Linear}(\overline{Y}^{(l)}) \tag{13}$$

$$Y^{(l)} = \text{LayerNorm}(\hat{Y}^{(l)}) \tag{14}$$

where we used Einstein summation notation to denote the tensor multiplications, MaskedSoftmax is a causally-masked softmax to prevent addressing future information, Linear is a linear layer applied per time-step and we omit reshaping operations for simplicity.

## C.2   RELATIVE MULTI-HEAD ATTENTION

The basic MHA operation does not take sequence order into account explicitly because it is permutation invariant, so positional encodings are a widely used solution in domains like language where order is an important semantic cue, appearing in the original transformer architecture (Vaswani et al., 2017). To enable a much larger contextual horizon than would otherwise be possible, we use the relative position encodings and memory scheme described in Dai et al. (2019). In this setting, there is an additional $\mathcal{T}$-step memory tensor $M^{(l)} \in \mathbb{R}^{\mathcal{T} \times D}$, which is treated as constant during weight updates.

**RelativeMultiHeadAttention**$(M^{(l-1)}, E^{(l-1)})$:

$$\widetilde{E}^{(l-1)} = [M^{(l-1)}, E^{(l-1)}] \tag{15}$$

$$Q^{(l)}, K^{(l)}, V^{(l)} = W_Q^{(l)} E^{(l-1)}, W_K^{(l)} \widetilde{E}^{(l-1)}, W_V^{(l)} \widetilde{E}^{(l-1)} \tag{16}$$

$$R = W_R^{(l)} \Phi \tag{17}$$

$$\alpha_{htm}^{(l)} = Q_{htd} K_{hmd} + Q_{htd} R_{hmd} + u_{h*d} K_{htm} + v_{h*d} R_{hmd} \tag{18}$$

$$W_{htm}^{(l)} = \text{MaskedSoftmax}(\alpha^{(l)}, \text{axis=}m) \tag{19}$$

$$\overline{Y}_{htd}^{(l)} = W_{htm}^{(l)} V_{hmd}^{(l)} \tag{20}$$

$$\hat{Y}^{(l)} = E^{(l-1)} + \text{Linear}(\overline{Y}^{(l)}) \tag{21}$$

$$Y^{(l)} = \text{LayerNorm}(\hat{Y}^{(l)}) \tag{22}$$

where $\Phi$ is the standard sinusoid encoding matrix, $u^{(l)}, v^{(l)} \in \mathbb{R}^{H \times d}$ are trainable parameters, the $*$ represents the broadcast operation, and $W_R$ is a linear projection used to produce the relative location-based keys (see Dai et al. (2019) for a detailed derivation).

### C.3 IDENTITY MAP REORDERING

The Identity Map Reordering modifies the standard transformer formulation as follows: the layer norm operations are applied only to the input of the sub-module and a non-linear ReLU activation is applied to the output stream.

$$\overline{Y}^{(l)} = \text{RelativeMultiHeadAttention}(\text{LayerNorm}([\text{StopGrad}(M^{(l-1)}), E^{(l-1)}])) \tag{23}$$

$$Y^{(l)} = E^{(l-1)} + \text{ReLU}(\overline{Y}^{(l)}) \tag{24}$$

$$\overline{E}^{(l)} = f^{(l)}(\text{LayerNorm}(Y^{(l)})) \tag{25}$$

$$E^{(l)} = Y^{(l)} + \text{ReLU}(\overline{E}^{(l)}) \tag{26}$$

See Figure 1 (Center) for a visual depiction of the TrXL-I.

## D  DMLAB-30 MEMORY/REACTIVE PARTITION

| Memory | Reactive |
|--------|----------|
| rooms_select_nonmatching_object | rooms_collect_good_objects_train |
| rooms_watermaze | rooms_exploit_deferred_effects_train |
| explore_obstructed_goals_small | rooms_keys_doors_puzzle |
| explore_goal_locations_small | language_select_described_object |
| explore_object_rewards_few | language_select_located_object |
| explore_obstructed_goals_large | language_execute_random_task |
| explore_goal_locations_large | language_answer_quantitative_question |
| explore_object_rewards_many | lasertag_one_opponent_large |
| | lasertag_three_opponents_large |
| | lasertag_one_opponent_small |
| | lasertag_three_opponents_small |
| | natlab_fixed_large_map |
| | natlab_varying_map_regrowth |
| | natlab_varying_map_randomized |
| | skymaze_irreversible_path_hard |
| | skymaze_irreversible_path_varied |
| | psychlab_arbitrary_visuomotor_mapping |
| | psychlab_continuous_recognition |
| | psychlab_sequential_comparison |
| | psychlab_visual_search |
| | explore_object_locations_small |
| | explore_object_locations_large |

Table 13: Partition of DMLab-30 levels into a memory-based and reactive split of levels.

| Model | Median Human Normalized Score |
|-------|-------------------------------|
| LSTM  | $136.6 \pm 3.4$ |
| GTrXL | $137.1 \pm 5.0$ |

Table 14: Final human-normalized median return across all 57 Atari levels for LSTM and GTrXL at 11.4 billion environment steps (equivalent to 200 million per individual game). Both models are 256 dimensions in width. We include standard error over runs.

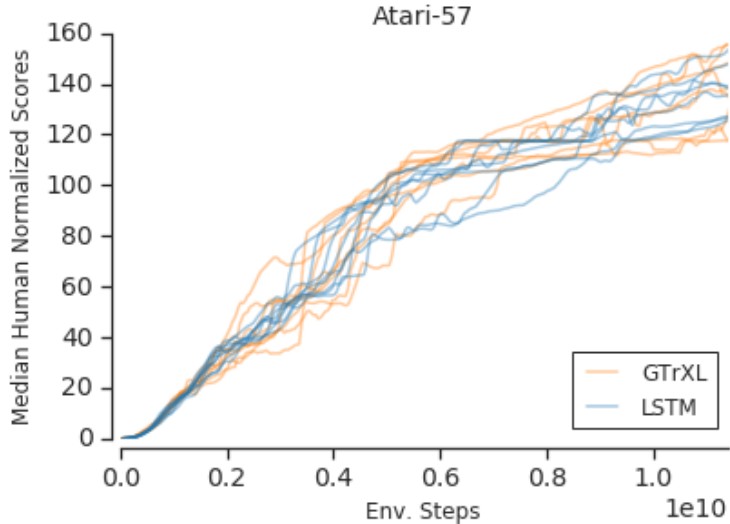

Figure 11: Median human-normalized returns as training progresses for both GTrXL and LSTM models. We run 8 hyperparameter settings per model.

# E    ATARI-57 RESULTS

In this section, we run the GTrXL on the multitask Atari-57 benchmark. Although Atari-57 was not designed specifically to test an agent's memory capabilities, we include these results here to demonstrate that we suffer no performance regression on a popular environment suite, providing further evidence that GTrXL can be used as an architectural replacement to the LSTM.

The LSTM and GTrXL are matched in width at 256 dimensions. The GTrXL is 12 layers deep to show our model's learning stability even at large capacity. The LSTM architecture matches the one reported in Anonymous Authors (2019). We train for 11.4 billion environment steps, equivalent to 200 million per environment. We run 8 hyperparameter settings per model.

