# OpenReview forum: "Stabilizing Transformers for Reinforcement Learning"
_ICLR.cc/2020/Conference — Reject_

### Official Review · AnonReviewer3 · 2019-10-21
**Official Blind Review #3**

**Rating:** 1

**Review:**

This paper is motivated by the unstable performance of Transformer in reinforcement learning, and tried several variants of Transformer to see whether some of them can stabilize the Transformer. The experimental results look good, however, I have problems in understanding the motivation, the intuition of the proposed methods, the experimental design, and the general implication to the research community that is using the Transformer in their day-to-day research.
First, the paper was based on the hypothesis of the authors that the Transformer is not stable, however, there is no comprehensive study on the unstability, and deep understanding on the root cause of it. It would be much more convincing to give a form definition of unstability and to add experimental study and theoretical analysis to the motivation part, instead of just based on a hypothesis.

Second, the proposal of the new structures (e.g., reordering the layer normalization, adding the gating layer) are quite ad hoc. There is not very solid motivation and theoretical analysis on why they could solve the unstable problem of the Transformer.  For example, by changing the order of layer normalization, there are direct identity mapping from the first layer to the last layer, making the information flow smoother. However, why this will make the Transformer more stable? The hypothesis and intuitive analysis are not very convincing. For another example, why replacing the residual connection with the gating layer can make the Transformer more stable? It seems to me that these are mostly heuristics, but not verified or strongly motivated solutions.

Third, if the proposal is sound, it should not be effective only for reinforcement learning. It should be able to improve the performance or stability of the Transformer in general (e.g., in NLP tasks). However, there is no experiments and discussions regarding this.

Fourth, the experiments on the reinforcement learning is a little narrow, and many famous RL benchmarks and environments were not tested. This makes it unclear whether the proposed approach is generally effective.


**I read the author responses, however, they do not really change my assessment on the paper.

**Experience Assessment:**

I have published in this field for several years.

**Review Assessment: Checking Correctness Of Derivations And Theory:**

I assessed the sensibility of the derivations and theory.

**Review Assessment: Checking Correctness Of Experiments:**

I carefully checked the experiments.

**Review Assessment: Thoroughness In Paper Reading:**

I read the paper at least twice and used my best judgement in assessing the paper.

---

> ### Author Response · Authors · 2019-11-11
> **Response to Review #3**
>
> Dear Reviewer 3,
>
> We address your criticisms below:
>
> > the paper was based on the hypothesis of the authors that the Transformer is not stable, however, there is no comprehensive study on the unstability
>
> The instability of the transformer architecture is demonstrated first-hand in our paper with the canonical variant failing to learn above a random policy’s performance in both DMLab-30 and the Memory Maze environment. These results were obtained over a large number of seeds and hyperparameter settings. Therefore the instability of the canonical transformer was not a hypothesis, it was repeatedly empirically demonstrated within our paper. We ask the reviewer to be more specific on what is insufficient about these results to make it not possible for us to claim that canonical transformers are unstable in RL domains. Beyond our own results, as referenced in the text, previous work [9] explicitly states a negative result when attempting to use a transformer for RL.
>
> > It would be much more convincing to give a form definition of unstability and to add experimental study and theoretical analysis to the motivation part, instead of just based on a hypothesis.
>
> Our paper is an empirically-driven study on using transformers in large-scale partially-observable RL environments, encompassing domains with high-dimensional observations and discrete, continuous and hybrid action spaces. Through a large number of experiments and ablations we propose a new gated variant of the transformer that is robust and easy to train (Sec. 4.3.2), and achieves the highest scores obtained so far on the DMLab-30 environment using less data than previous methods (Sec. 4.1) [1][2][3][4][5]. Our results are statistically significant, obtained over several seeds and hyperparameter settings. We believe it is unreasonable to discard these extensive empirical results only because there isn’t a strict theoretical understanding of the architecture.
>
> > It seems to me that these are mostly heuristics, but not verified or strongly motivated solutions.
>
> It is not clear that an improved architecture design needs theoretical insight as a prerequisite to being useful to the field, as even the original transformer paper does not present substantial theoretical results. Previous work which applied gating / multiplicative interactions in feedforward architectures and demonstrated significant empirical improvements had widespread impact to their respective fields [6][7][9][10]. With respect to these architectural modifications being “not verified”, we request the reviewer to be more specific about what is missing in our experimental results.

---

> > ### Author Response · Authors · 2019-11-11
> > **Response to Review #3 cont**
> >
> > > It should be able to improve the performance or stability of the Transformer in general (e.g., in NLP tasks)
> >
> > This work explores the use of the transformer as a memory architecture for reinforcement learning agents and we accordingly limited the experiments to partially-observable reinforcement learning domains. We believe that supervised and reinforcement learning domains are sufficiently distinct (e.g. for RL: learning rate schedules being more difficult to tune, no regularization, gradient clipping having a larger effect, higher variance of the objective function, etc) and that focusing on only one of the domains is valid. We leave applying the gated transformer to supervised learning settings to future work.
> >
> > > the experiments on the reinforcement learning is a little narrow, and many famous RL benchmarks and environments were not tested
> >
> > Beyond DMLab-30, there are a very limited number of partially-observable reinforcement learning benchmarks where memory plays a critical role in performance, and that none of these reach the difficulty of multitask DMLab-30. Standard benchmarks such as Atari-57 and the Mujoco OpenAI gym environments do not make extensive use of memory (or are even fully observable), and applying the transformer there would not be informative. For further proof, we have run the GTrXL on Atari-57 and, as expected from previous work, the transformer’s memory provides no substantial gain over a baseline LSTM agent.
> >
> > Atari-57:
> > Model  | Median Human Normalized Score
> > LSTM  | 136.6 +- 3.4
> > GTrXL | 137.1 +- 5.0
> >
> >
> > [1] Recurrent Experience Replay in Distributed Reinforcement Learning. Steven Kapturowski, Georg Ostrovski, Will Dabney, John Quan, Remi Munos. International Conference on Learning Representations. 2019.
> > [2] Impala: Scalable distributed deep-rl with importance weighted actor-learner architectures. Lasse Espeholt, Hubert Soyer, Remi Munos, Karen Simonyan, Volodymir Mnih, Tom Ward, Yotam Doron, Vlad Firoiu, Tim Harley, Iain Dunning, Shane Legg, Koray Kavukcuoglu. International Conference on Machine Learning. 2018.
> > [3] Multi-task Deep Reinforcement Learning with PopArt. Matteo Hessel, Hubert Soyer, Lasse Espeholt, Wojciech Czarnecki, Simon Schmitt, Hado van Hasselt. Association for the Advancement of Artificial Intelligence. 2018.
> > [4] Information Asymmetry in KL-regularized RL. Alexandre Galashov, Siddhant M. Jayakumar, Leonard Hasenclever, Dhruva Tirumala, Jonathan Schwarz, Guillaume Desjardins, Wojciech M. Czarnecki, Yee Whye Teh, Razvan Pascanu, Nicolas Heess. International Conference on Learning Representations. 2019.
> > [5] Mix & Match – Agent Curricula for Reinforcement Learning. Wojciech Marian Czarnecki, Siddhant M. Jayakumar, Max Jaderberg, Leonard Hasenclever, Yee Whye Teh, Simon Osindero, Nicolas Heess, Razvan Pascanu. International Conference on Machine Learning. 2018.
> > [6] Highway Networks. Rupesh Kumar Srivastava, Klaus Greff, Jürgen Schmidhuber. 2015.
> > [7] Language Modeling with Gated Convolutional Networks. Yann N. Dauphin, Angela Fan, Michael Auli, David Grangier. International Conference on Machine Learning. 2017.
> > [8] A Simple Neural Attentive Meta-Learner. Nikhil Mishra, Mostafa Rohaninejad, Xi Chen, Pieter Abbeel. International Conference on Learning Representations. 2018.
> > [9] Squeeze-and-Excitation Networks. Jie Hu, Li Shen, Samuel Albanie, Gang Sun, Enhua Wu. IEEE conference on computer vision and pattern recognition. 2018.
> > [10] Conditional Image Generation with PixelCNN Decoders. Aaron van den Oord, Nal Kalchbrenner, Oriol Vinyals, Lasse Espeholt, Alex Graves, Koray Kavukcuoglu. 2016.

---

### Official Review · AnonReviewer1 · 2019-10-24
**Official Blind Review #1**

**Rating:** 3

**Review:**

The paper explores a transformer for reinforcement learning. The authors demonstrate that Canonical Transformer is unstable. The authors introduce two modifications to the Canonical Transformer. The first is to move the layer normalization layer to the input stream. The second is to replace residual connections with gating layers. The experimental results show that (1) the first modification, i.e., moving the layer normalization layer to the input stream significantly stabilizes the training; (2) Gated Recurrent Unit (GRU) gating seems to be most effective gating mechanism.
My decision is Weak Reject, considering the following aspects.

Positive points: (1) The experiments seem solid. The authors have evaluated the overall performance, as well as hyperparameters, seeds, and ablations. (2) Moving the layer normalization layer to the input stream seems to be surprisingly effective. This could be an interesting finding. (3) The paper is well organized.

Negative points: (1) Lack of experiments on benchmark and large environments. The authors did not evaluate their model on the widely used benchmark Atari-57. Also, it is unclear whether the proposed transformer can scale to large environments. (2) Lack of understanding of the layer normalization. The authors provide some explanations about why the reordering works, but they seem not intuitive. More analysis about why the reordering works would significantly enhance this paper.

Specific questions: (1) Have you tried simply removing the layer normalization layer? (2) TrXL-I moves two layer-normalization layers together. Have you tried only moving one of them? Which modification contributes more? (3) Could you provide more explanations about why the modification of the layer normalization layer works? (4) Have you experimentally validated the proposed hypothesis as to why the Identity Map Reordering, such as recording the evolution of the produced values in the submodules?


**Experience Assessment:**

I have published one or two papers in this area.

**Review Assessment: Checking Correctness Of Derivations And Theory:**

N/A

**Review Assessment: Checking Correctness Of Experiments:**

I carefully checked the experiments.

**Review Assessment: Thoroughness In Paper Reading:**

I read the paper thoroughly.

---

> ### Author Response · Authors · 2019-11-11
> **Response to Review #1**
>
> Dear Reviewer 1,
>
> We thank you for your comments and constructive criticism. We hope to address your concerns below:
>
> > Lack of experiments on benchmark and large environments... it is unclear whether the proposed transformer can scale to large environments
>
> We want to highlight that DMLab-30 is a large-scale, multitask benchmark arguably more challenging than Atari-57, comprising 30 first-person, 3D environments with image observations. The tasks require a range of agent competencies such as language comprehension, navigation, handling of partial observability, memory, planning, and other forms of long horizon reasoning, with episodes lasting over 4000 steps. It has been used in a large number of previous works, see e.g. [1][2][3][4][5] for a subset, with the previous multitask state-of-the-art being [3]. With the substantial amount of previous work that went into obtaining high scores on the DMLab-30 suite as context, the GTrXL results we report in this paper are the highest so far obtained for this benchmark.
>
> > The authors did not evaluate their model on the widely used benchmark Atari-57.
>
> The transformer is mainly a memory architecture and previous work has shown minimal improvement from using memory on Atari-57 (see e.g.in RND [6], on Montezuma’s Revenge a feedforward CNN achieves scores close to a recurrent model). While the Atari-57 tasks themselves might not always be fully observable, they are not meant to test the memory of the agent. On the other hand, DMLab-30 was explicitly designed to require memory in order to solve it.
>
> Despite this, we have run the GTrXL on the Atari-57 suite. As in previous work, memory does not seem to play a large part in performance and there is no substantial improvement of the transformer over a baseline LSTM architecture. However, we can at least note that the GTrXL suffers no performance degradation on this set of environments, and that it shows stability comparable to an LSTM.
>
> Atari-57:
> Model  | Median Human Normalized Score
> LSTM  | 136.6 +- 3.4
> GTrXL | 137.1 +- 5.0

---

> > ### Author Response · Authors · 2019-11-11
> > **Response to Review #1 cont**
> >
> > > Lack of understanding of the layer normalization. The authors provide some explanations about why the reordering works, but they seem not intuitive. More analysis about why the reordering works would significantly enhance this paper.
> >
> > We first want to reiterate that reordering the layer norm is not an original contribution of our work, as we cited in our paper it has been established in transformers previously [7][8] and earlier for convolutional networks [9]. A definitive reason for keeping the residual stream untransformed through depth still remains an open question. Our work empirically demonstrates that it provides substantial improvements in stability for RL domains which we believe, by itself, is an important contribution for others working in the field. See our answer to point (3-4) for a summary of a previously established hypothesis for why reordering normalization works.
> >
> > >Have you tried simply removing the layer normalization layer?
> > An initial experiment removing layer norm performed substantially worse, even on a shallow 4-layer transformer.
> >
> > > TrXL-I moves two layer-normalization layers together. Have you tried only moving one of them? Which modification contributes more?
> > While this is definitely an interesting question, it is a convention to place the layer norm at the same place and due to the already substantial amount of computation time the ablations required, we were not able to test this.
> >
> > > Specific Points (3-4)
> > We believe our reasoning for reordering the layer norm operator largely follows the same lines as previous work examining normalization placement (see section 2 of [9]). Having an identity mapping (the skip connection in the residual connection) enables gradients to flow from any layer l to any shallower layer k<l untransformed, allowing an easier optimization as e.g. gradients can flow from the policy and value head outputs directly to the image encoder without any intermediate transformation. If the layer norm was placed after residual recombination, then during backpropagation the gradients at a shallower layer k<l are non-linearly transformed by the backwards pass of each layer m’s layer norm, where k<m<l.
> >
> > [1] Recurrent Experience Replay in Distributed Reinforcement Learning. Steven Kapturowski, Georg Ostrovski, Will Dabney, John Quan, Remi Munos. International Conference on Learning Representations. 2019.
> > [2] Impala: Scalable distributed deep-rl with importance weighted actor-learner architectures. Lasse Espeholt, Hubert Soyer, Remi Munos, Karen Simonyan, Volodymir Mnih, Tom Ward, Yotam Doron, Vlad Firoiu, Tim Harley, Iain Dunning, Shane Legg, Koray Kavukcuoglu. International Conference on Machine Learning. 2018.
> > [3] Multi-task Deep Reinforcement Learning with PopArt. Matteo Hessel, Hubert Soyer, Lasse Espeholt, Wojciech Czarnecki, Simon Schmitt, Hado van Hasselt. Association for the Advancement of Artificial Intelligence. 2018.
> > [4] Information Asymmetry in KL-regularized RL. Alexandre Galashov, Siddhant M. Jayakumar, Leonard Hasenclever, Dhruva Tirumala, Jonathan Schwarz, Guillaume Desjardins, Wojciech M. Czarnecki, Yee Whye Teh, Razvan Pascanu, Nicolas Heess. International Conference on Learning Representations. 2019.
> > [5] Mix & Match – Agent Curricula for Reinforcement Learning. Wojciech Marian Czarnecki, Siddhant M. Jayakumar, Max Jaderberg, Leonard Hasenclever, Yee Whye Teh, Simon Osindero, Nicolas Heess, Razvan Pascanu. International Conference on Machine Learning. 2018.
> > [6] Exploration by Random Network Distillation. Yuri Burda, Harrison Edwards, Amos Storkey, Oleg Klimov. International Conference on Learning Representations. 2019.
> > [7] Adaptive Input Representations for Neural Language Modeling. Alexei Baevski, Michael Auli. International Conference on Learning Representations. 2019.
> > [8] Language Models are Unsupervised Multitask Learners. Alec Radford, Jeffrey Wu, Rewon Child, David Luan, Dario Amodei, Ilya Sutskever. 2019.
> > [9] Identity Mappings in Deep Residual Networks. Kaiming He, Xiangyu Zhang, Shaoqing Ren, Jian Sun. European conference on computer vision. 2016.

---

### Official Review · AnonReviewer4 · 2019-11-08
**Official Blind Review #4**

**Rating:** 3

**Review:**

* Summary
This paper introduces architecture modifications for self-attention to stabilize transformers in reinforcement learning.
The new architecture, Gated Transformer-XL, replaces the order of the layer norm blocks to preserve an identity mapping.
Multiple existing gating layers are proposed to replace the residual connections of transformer.
The new architecture is compared against MERLIN and LSTM on DMlab-30, and further ablation studies are done on Numpad and Memory Maze.
It is noted that they use the recent V-MPO objective to train LSTM and transformer.
Their results show that transformers are able to learn in memory-intensive environments, with some gating combinations surpassing LSTM.

* Decision
This paper presents promising empirical results, however the experiments are limited, making it difficult to place in the broader work.
In addition, the contribution is incremental and not well-motivated.
I would recommend a weak rejection.
Still, I think the paper is well written and could be improved upon.

* Reasons
While the empirical results are impressive, they are not put into context.
DMlab-30 is still a relatively new environment suite and it is difficult to place the result of this paper in the context of broader work.
In addition, the comparison is against LSTM on a new objective.
While Transformer is able to beat LSTM on the same objective, it is unclear whether that is a success of the objective or the architecture.
In Numpad, the transformer architecture shows an improvement over LSTM, but no comparisons are made to any other memory-based agents.
The hyperparameter studies on Memory Maze also show improvements in memory-related tasks, but do not help in understanding of the proposed work.

The choice of architecture modification is also not well motivated.
As the paper mentions, initializing near identity has been shown to be important in the supervised learning literature.
For the topic of this paper however, I do not think this adequetly explains the instability of transformers in reinforcement learning.
In the related work section for example, the paper notes that gating mechanisms have been used to handle the vanishing gradients problem.
The paper also notes that vanishing gradients is not an issue in transformers.
Hence, it is unclear why gating would stabilize transformers for reinforcement learning.

The paper is overall well written and the ideas developed are clear.
Unfortunately, the impressive results on DMlab are not sufficient for both the lack of deeper empirical study and better theoretical motivation for the architecture modifications.

**Experience Assessment:**

I have read many papers in this area.

**Review Assessment: Checking Correctness Of Derivations And Theory:**

N/A

**Review Assessment: Checking Correctness Of Experiments:**

I carefully checked the experiments.

**Review Assessment: Thoroughness In Paper Reading:**

I read the paper at least twice and used my best judgement in assessing the paper.

---

> ### Author Response · Authors · 2019-11-11
> **Response to Review #4**
>
> Dear Reviewer 4,
>
> Thank you for your comments and feedback. We hope to address your concerns below:
>
> >DMlab-30 is still a relatively new environment suite and it is difficult to place the result of this paper in the context of broader work.
>
> We will make more effort to better contextualize the DMLab-30 results that we reported and intend to expand this exposition in the next version of the paper. To summarize: (1) there has been significant previous work on improving scores on DMLab-30 (and subsets of the 30 levels), see e.g. [1][2][3][4][5] for a few works, and (2) our proposed architecture achieves the best scores so far reported on this benchmark and with better data efficiency, even reporting scores better than previous results where different models were trained on each of the 30 levels [1][2].
>
> > In addition, the comparison is against LSTM on a new objective. While Transformer is able to beat LSTM on the same objective, it is unclear whether that is a success of the objective or the architecture.
>
> We want to highlight that the LSTM trained with the V-MPO objective is state-of-the-art for LSTM architectures on DMLab-30 over all RL algorithms reported so far, and we believe this provides evidence that the objective does not somehow unfairly bias improved performance only in transformer architectures.
>
> > In Numpad, the transformer architecture shows an improvement over LSTM, but no comparisons are made to any other memory-based agents.
>
> The toy numpad environment was mainly to highlight that our model works much better than an LSTM even within partially-observable continuous control applications.
>
> > adequate theoretical understanding of architecture
>
> Our goal in this paper was not a theoretical understanding of the optimization stability of transformers and we make no claim towards that goal. Our goal was instead to (1) demonstrate that there is significant empirical instability when the canonical transformer is used in RL applications, and (2) propose a set of modifications which seem to alleviate instability, validated in a large-scale rigorous empirical study. We believe these results will be valuable to others who want to exploit the powerful memory capability of transformers in RL. Previous work that applied gating to feedforward architectures had a substantial impact on their respective fields without a formal theoretical understanding of why gating helps [6][7][8][9].
>
> > … the paper notes that gating mechanisms have been used to handle the vanishing gradients problem. The paper also notes that vanishing gradients is not an issue in transformers.
>
> We believe the transformer’s recent success on long-range modeling of temporal dependencies provides evidence that vanishing gradients are not an issue for transformers along the time axis. The gating mechanism we applied was along the depth axis, not time, and therefore we believe our statements were not in direct contradiction.
>
> [1] Recurrent Experience Replay in Distributed Reinforcement Learning. Steven Kapturowski, Georg Ostrovski, Will Dabney, John Quan, Remi Munos. International Conference on Learning Representations/ 2019.
> [2] Impala: Scalable distributed deep-rl with importance weighted actor-learner architectures.
> Lasse Espeholt, Hubert Soyer, Remi Munos, Karen Simonyan, Volodymir Mnih, Tom Ward, Yotam Doron, Vlad Firoiu, Tim Harley, Iain Dunning, Shane Legg, Koray Kavukcuoglu. International Conference on Machine Learning. 2018.
> [3] Multi-task Deep Reinforcement Learning with PopArt. Matteo Hessel, Hubert Soyer, Lasse Espeholt, Wojciech Czarnecki, Simon Schmitt, Hado van Hasselt. Association for the Advancement of Artificial Intelligence. 2018.
> [4] Information Asymmetry in KL-regularized RL. Alexandre Galashov, Siddhant M. Jayakumar, Leonard Hasenclever, Dhruva Tirumala, Jonathan Schwarz, Guillaume Desjardins, Wojciech M. Czarnecki, Yee Whye Teh, Razvan Pascanu, Nicolas Heess. International Conference on Learning Representations. 2019.
> [5] Mix & Match – Agent Curricula for Reinforcement Learning. Wojciech Marian Czarnecki, Siddhant M. Jayakumar, Max Jaderberg, Leonard Hasenclever, Yee Whye Teh, Simon Osindero, Nicolas Heess, Razvan Pascanu. International Conference on Machine Learning. 2018.
> [6] Language Modeling with Gated Convolutional Networks. Yann N. Dauphin, Angela Fan, Michael Auli, David Grangier. International Conference on Machine Learning. 2017.
> [7] Squeeze-and-Excitation Networks. Jie Hu, Li Shen, Samuel Albanie, Gang Sun, Enhua Wu. IEEE conference on computer vision, pattern recognition. 2018.
> [8] Conditional Image Generation with PixelCNN Decoders. Aaron van den Oord, Nal Kalchbrenner, Oriol Vinyals, Lasse Espeholt, Alex Graves, Koray Kavukcuoglu. 2016.
> [9] Highway Networks. Rupesh Kumar Srivastava, Klaus Greff, Jürgen Schmidhuber. 2015.

---

### Public Comment · ~julien_perez1 · 2019-10-18
**possible related reference**

Thanks for this paper, I'm sharing a possible related work in the context of language comprehension and dialog:
https://www.aclweb.org/anthology/E17-1001/

Regards,

---

### Author Response · Authors · 2019-11-11
**Paper Revision**

Dear Reviewers,

We have uploaded a paper revision that we hope addresses some of the feedback made in reviews. We have expanded the section describing DMLab-30 with references to previous work that used the 30 levels or subsets of them as benchmark environments for algorithms and architectures. Additionally, as requested by some reviewers we include in the appendix additional results on the multitask Atari-57 benchmark. We see no improvements over an LSTM baseline, which we believe is expected as Atari-57 is not meant to test an agent's memory capabilities.

---

### Author Response · Authors · 2019-11-14
**Paper Revision #2**

Dear Reviewers,

We have submitted another revision where in Section 4.3.3 we include the results of a substantially larger LSTM baseline that is parameter-matched to the transformers we trained. This model performs only marginally better than the original baseline LSTM and the performance gains are mainly in the reactive set of levels.

---

### Public Comment · ~Marco_Pleines1 · 2022-04-19
**Lack of reproducibility due to missing source code and technical details**

Adding a Transformer architecture to Deep Reinforcement Learning algorithms is a non-trivial process. It is great that a few people, like the authors and Lampinen et al. 2021, got it working while showing promising results towards multiple POMDPs. However, there is one major concern. There is no clear implementation available. Without access to the source code, many important implementations details are missing that ensure reproducibility. It is unclear what kind of data is stored in the episodic memory (raw observations? encoded features?) and how the interface between the memory and the training algorithm works. There are likely more questions that the authors should address to make their research accessible and hence reproducible, which should be the default for scientific work. This issue is also apparent in the published ICML paper.

---

### Decision · Program_Chairs · 2019-12-19

**Decision:**

Reject

**Comment:**

This paper proposes  architectural modifications to transformers, which are promising for sequential tasks requiring memory but can be unstable to optimize, and applies the resulting method to the RL setting, evaluated in the DMLab-30 benchmark.

While I thought the approach was interesting and the results promising, the reviewers unanimously felt that the experimental evaluation could be more thorough, and were concerned with the motivation behind of some of the proposed changes.